# Expanding Roles of De Novo Lipogenesis in Breast Cancer

**DOI:** 10.3390/ijerph18073575

**Published:** 2021-03-30

**Authors:** Pasquale Simeone, Stefano Tacconi, Serena Longo, Paola Lanuti, Sara Bravaccini, Francesca Pirini, Sara Ravaioli, Luciana Dini, Anna M. Giudetti

**Affiliations:** 1Department of Medicine and Aging Sciences, University “G. d’Annunzio”, Chieti-Pescara, 66100 Chieti, Italy; simeone.pasquale@gmail.com (P.S.); paola.lanuti@unich.it (P.L.); 2Center for Advanced Studies and Technology (CAST), University “G. d’Annunzio”, Chieti-Pescara, 66100 Chieti, Italy; 3Department of Biological and Environmental Sciences and Technologies, University of Salento, Via Prov.le Lecce-Monteroni, 73100 Lecce, Italy; stefano.tacconi@unisalento.it (S.T.); serena.longo@unisalento.it (S.L.); 4IRCCS Istituto Romagnolo per lo Studio dei Tumori (IRST) “Dino Amadori”, 47014 Meldola, Italy; sara.bravaccini@irst.emr.it (S.B.); francesca.pirini@irst.emr.it (F.P.); sara.ravaioli@irst.emr.it (S.R.); 5Department of Biology and Biotechnology “C. Darwin”, Sapienza University of Rome, 00185 Rome, Italy; luciana.dini@uniroma1.it; 6CNR Nanotec, 73100 Lecce, Italy

**Keywords:** breast cancer, de novo lipogenesis, extracellular vesicles, metabolism

## Abstract

In recent years, lipid metabolism has gained greater attention in several diseases including cancer. Dysregulation of fatty acid metabolism is a key component in breast cancer malignant transformation. In particular, de novo lipogenesis provides the substrate required by the proliferating tumor cells to maintain their membrane composition and energetic functions during enhanced growth. However, it appears that not all breast cancer subtypes depend on de novo lipogenesis for fatty acid replenishment. Indeed, while breast cancer luminal subtypes rely on de novo lipogenesis, the basal-like receptor-negative subtype overexpresses genes involved in the utilization of exogenous-derived fatty acids, in the synthesis of triacylglycerols and lipid droplets, and fatty acid oxidation. These metabolic differences are specifically associated with genomic and proteomic changes that can perturb lipogenic enzymes and related pathways. This behavior is further supported by the observation that breast cancer patients can be stratified according to their molecular profiles. Moreover, the discovery that extracellular vesicles act as a vehicle of metabolic enzymes and oncometabolites may provide the opportunity to noninvasively define tumor metabolic signature. Here, we focus on de novo lipogenesis and the specific differences exhibited by breast cancer subtypes and examine the functional contribution of lipogenic enzymes and associated transcription factors in the regulation of tumorigenic processes.

## 1. Introduction

Breast cancer is the most common cancer in female individuals. In 2020, breast, lung, and colorectal accounted for 50% of all new diagnoses of female cancers, and breast cancer alone accounts for 30% [1]. Breast cancer is a highly heterogeneous disease characterized by the presence of different subtypes with variable clinical outcomes. With the growing amount of data generated by proteogenomic studies, researchers have been able to define a detailed molecular landscape of each subtype, thus providing a detailed picture of genomic alterations and altered functional networks [2,3,4].

Transcriptional profiling studies have defined four major breast cancer subtypes: luminal A, luminal B, HER2-enriched, and basal-like [5]. Luminal breast cancers, which include luminal subtypes A and B, are characterized by estrogen receptor (ER) and/or progesterone (PR) expression and confers a more favorable prognosis in part due to reactivity to anti-hormone therapies [6]. In contrast, baseline-type breast cancer such as ER negative, PR negative, and HER2 negative (triple negative breast cancer, TNBC) has the highest recurrence rate and worst overall survival rate among all breast cancer subtypes [6]. As a consequence of this different genomic and proteomic background, breast cancers can also be classified in terms of altered metabolic pathways and metabolite levels. For instance, from the analysis of 228 non-treated breast cancer patients, three significantly different metabolic clusters (Mc1, Mc2, and Mc3) have emerged. Mc1 is characterized by the highest levels of glycerophosphocholine and phosphocholine, Mc2 by the highest levels of glucose, and Mc3 by the highest levels of lactate and alanine [7].

This metabolic phenotyping results in enhanced nutrient uptake to support the metabolic demand of tumor cells, but also to provide cells with metabolic intermediates, named oncometabolites, which sustain specific cellular processes [8,9]. For instance, increased lactate production is a characteristic of ER^−^ tumors due to the increased rate of glycolysis and over-expression of lactate dehydrogenases [10].

In this scenario, lipid metabolic reprogramming is a hallmark in cancer [10,11]. Cancer cells exhibit a “lipogenic phenotype” characterized by exacerbated levels of fatty acid biogenesis, even in the presence of abundant circulating exogenous fatty acids and reflected in the overexpression and increased activity of lipogenic enzymes [11,12,13,14]. Fatty acids or derived lipid species are then critical for lipid synthesis and membrane structures, protein modification and localization functions, and receptor localization and signaling of major oncogenic signaling pathways [15].

Accordingly, the uptake of preformed fatty acids may also be an important mechanism of cancer lipid acquisition. Several factors including genetic mutations and metabolic conditions of oxygen and nutrient deprivation can drive this metabolic adaptation [16]. In the context of breast cancer, while the luminal subtype relies on de novo lipogenesis (*DN*L) (or de novo fatty acid synthesis) as a source for biomass and energy requirements, basal-like TNBC cells overexpress genes involved in the utilization of exogenous fatty acids and triacylglycerol synthesis [17]. Lipid transport and uptake are indeed important and under-appreciated aspects of lipid metabolism in cancer [18]. This aspect links lipid metabolism with the activation of particular tumor cellular processes such as the epithelial–mesenchymal transition (EMT) that is thought to contribute to cancer progression and chemoresistance [19,20,21].

These findings highlight the role of lipid metabolism in breast cancer pathophysiology and the tight correlation with multiple cellular processes. Here, we review this functional crosstalk focusing on metabolic pathways related to *DN*L.

## 2. Lipogenesis in Cancer

In normal tissues, glucose represents the main source of acetyl-CoA for lipid synthesis, a process known as *DN*L (Figure 1).

In this case, pyruvate, the glycolytic end-product, enters mitochondria where it is converted in acetyl-CoA by the pyruvate dehydrogenase complex. In hypoxia, a common feature of solid tumors, a metabolic reprogramming accounts for a reduction in the mitochondrial conversion of pyruvate into acetyl-CoA that normally feeds the tricarboxylic acid cycle (TCA) to produce citrate. As a result, glycolysis-derived pyruvate is primarily diverted to lactate rather than to mitochondrial oxidative phosphorylation [22]. This metabolic reprogramming allows the cancer cells to rapidly obtain ATP and glycolytic intermediates to support anabolic demand.

Cancer cells adopt different metabolic adaptations for fatty acid synthesis, mainly relying on glutamine or acetate as alternative substrates [22]. The reductive carboxylation of glutamine-derived α-ketoglutarate, through the isocitrate dehydrogenase-1 (IDH1)-dependent pathway represents, for different cancers, the main pathway to synthesize acetyl-CoA for lipid synthesis [23]. TNBC cells often exhibit glutamine-dependent phenotype upregulating both glutamine uptake and glutamine-related enzymes [24,25]. Consequently, TNBC may be more susceptible to glutamine-targeting therapeutics compared to luminal types [25].

Acetate can be derived from a range of sources including extracellular acetate, histone deacetylation, and the recently identified conversion of pyruvate into acetate by thiamine-dependent keto acid dehydrogenases as well as a reactive oxygen species-coupled reaction [26].

In hypoxic conditions, breast cancers increase acetate uptake [27,28,29,30] and upregulate enzymes converting acetate to acetyl-CoA such as acetyl-CoA synthetases (ACSS). Indeed, acyl-CoA synthetase short-chain family member 2 (ACSS2) was found to be upregulated in hypoxic breast cells [27,30].

Hypoxic cells can also rely on the uptake of fatty acids to compensate for reduced glucose-based *DN*L [26,31]. Breast cancers have increased expression of fatty acid-binding proteins (FABP3, FABP7 or FABP4) and CD36 [32,33], which are involved in the uptake and subcellular trafficking of fatty acids [31,34,35]. Thus, in breast cancer cells, lipid accumulation in the form of lipid droplets relies on a FABP-dependent fatty acid uptake, while *DN*L resulted in being repressed. Indeed, CD36 inhibition impaired angiogenesis as well as migration and invasion of breast cancer cell lines [36,37].

## 3. Key Enzymes of De Novo Lipogenesis (*DN*L)

Cell requirement for fatty acids is normally met by the utilization of dietary fatty acids. The amount of fatty acids synthetizes by *DN*L is of minor importance in most human tissues, except for the liver, mammary gland, and to a lesser extent, adipose tissue [38]. However, the rate of *DN*L and the expression of several lipogenic enzymes are increased in various cancer types.

*DN*L is a metabolic process by which pyruvate, mainly derived from carbohydrate sources is converted into fatty acids (Figure 1) [38].

In aerobic conditions, pyruvate, the end product of glycolysis is transformed in acetyl-CoA, which enters the TCA cycle by condensing with oxaloacetate to form citrate. When the energetic charge is high such as after a meal rich in carbohydrates, citrate can be transported from the mitochondria into the cytosol, where fatty acid synthesis occurs. Citrate efflux into the cytosol is catalyzed by the citrate carrier (CIC), an intrinsic protein of the inner mitochondrial membrane, which catalyzes an electroneutral exchange of citrate plus a proton with malate. In the cytosol, glucose-derived citrate is converted into oxaloacetate and acetyl-CoA by the ATP citrate lyase (ACLY). The obtained acetyl-CoA is required for lipid synthesis during membrane biogenesis as well as for histone acetylation reactions to regulate the expression of certain proteins in aberrantly proliferating cancer cells.

Key enzymes of *DN*L are acetyl-CoA carboxylase (ACC) and the multi enzymatic complex fatty acid synthase (FASN).

ACC represents the rate-limiting enzyme of *DN*L, catalyzing the irreversible carboxylation of acetyl-CoA into malonyl-CoA. The reaction requires biotin and ATP. In humans and mammals, there are two isoforms of ACCs: ACC1 (or ACC-α) with 265 kDa and ACC2 (or ACC-β) with 280 kDa. ACC1 presents in the cytosol with a lipogenic role, so is particularly expressed in lipogenic tissues and ACC2 is mainly associated with mitochondria in oxidative tissues [39,40,41]. Therefore, ACC1 is enriched in lipogenic tissues such as the liver, adipose, and lactating mammary gland, where it catalyzes the biosynthesis of long-chain fatty acids. In contrast, ACC2 is highly expressed in oxidative tissues such as skeletal muscle and heart, where it regulates fatty acid β-oxidation.

Cytosolic malonyl-CoA, produced by ACC, can be used for fatty acid biosynthesis. The reaction is catalyzed by FASN. After priming with acetyl-CoA, FASN uses malonyl-CoA as a carbon donor and NADPH as a reduced cofactor to produce palmitoyl-CoA.

By furnishing malonyl-CoA, ACC not only plays a key role in *DN*L, but also regulates mitochondrial fatty acid β-oxidation, considering that malonyl-CoA is an inhibitor of carnitine palmitoyl-transferase-1, the key enzyme of this metabolic process.

The de novo synthesized fatty acids can be used for the synthesis of complex lipids such as phospholipids, ceramides, cholesterol esters, and triacylglycerols and thereby play a major role in membrane structure, cell signaling, and energy storage. Following *DN*L, the enzyme stearoyl-CoA desaturase (SCD) catalyzes the introduction of the first double bond in the cis-delta 9 position of saturated fatty acyl-CoA giving monounsaturated fatty acids, which are preferentially transformed into triacylglycerols for storage [41]. Two isoforms of SCD have been reported in human cells, SCD1 and SCD5 [42]. Both isoforms are overexpressed in luminal cancer models compared to the TNBC subtypes [43].

## 4. Transcription Factors Regulating De Novo Lipogenesis (*DN*L)

*DN*L is a highly regulated metabolic pathway. Having common features at their promoter regions, lipogenic genes are coordinately regulated at the transcription level. The transcription factors sterol regulatory element-binding protein-1c (SREBP-1c), upstream stimulatory factor (USF), peroxisome-proliferation-activated receptors (PPARs), carbohydrate response element-binding protein (ChREBP), and liver X receptors (LXRs) play critical roles in regulating this process (Figure 1).

SREBPs represent the master transcriptional factors regulating *DN*L. SREBPs are members of the basic helix-loop-helix (bHLH)-leucine zipper transcription factors and can be classified into three types: SREBP-2, SREBP-1a, and SREBP-1c. Whereas SREBP-2 preferentially regulates genes involved in cholesterol metabolism, SREBP-1 regulates fatty acid synthesis enzymes. Expressions of ACC, FASN, and SCD1 are under the control of SREBP-1c [44]. Moreover, SREBP-1c activates the expression of CIC both in hepatocytes [45] and in the mammary epithelium [46] and SREBP-1 overexpression increases the CIC transcript and protein levels. Moreover, SREBP-1 upregulates ACLY at the mRNA level via Akt signaling [47].

Although SREBP-1 plays a pivotal role in regulating lipogenic gene expression, it is not the only one. In vitro studies have demonstrated that insulin effect on FASN promoter also requires the presence of the upstream stimulatory factors (USFs). USFs are bHLH-leucine zipper transcription factors able to bind the CANNTG sequence present in the promoter region of FASN. The effects of SREBP-1 and USFs on FASN are independent and additive [48].

Lipogenic enzyme transcription may also be regulated by ChREBP [49], a glucose-regulated bHLH transcription factor. In response to increased glucose levels, ChREBP undergoes dephosphorylation steps that allow translocation from the cytoplasm to the nucleus where, in association with its binding partner Max-like (MLX) interacting protein, it binds carbohydrate response elements of lipogenic genes [50,51,52].

LXRs are members of the nuclear receptor superfamily that heterodimerize with retinoid X receptor (RXR) [53]. Two isoforms of LXRs have been identified, LXRα and LXRβ [53,54]. It has been reported that LXRs perform an important role in the regulation of fatty acid synthesis. LXRs can activate lipogenic enzymes directly or by SREBP-1c. FASN is transcriptionally regulated by both LXRα and LXRβ [54,55,56].

PPARs are members of the superfamily of nuclear hormone receptors that function as ligand-dependent transcription factors. Upon ligand activation, they regulate the expression of genes containing a specific response element, called the PPAR-responsive element (PPRE), which consists of a hexameric nucleotide direct repeat of the recognition motif (TGACCT) spaced by one nucleotide (DR-1). Three subtypes of PPARs termed α, δ (or β), and γ, have been identified [57,58]. These receptors heterodimerize with the retinoid X receptor (RXR) and alter the transcription of target genes after binding to PPRE.

Although PPARγ is considered to be the master regulator of adipocyte differentiation, an increase in PPARγ expression has been associated with accumulation in hepatic triacylglycerols. A study reports that PPARγ is capable of inducing lipid accumulation in hepatocytes in which an increase in SREBP-1 as well as ACC and FASN expression is also measured. These data suggest that PPARγ may play a role in stimulating lipogenesis [57,58,59]. Heterozygous PPARγ mutant mice exhibit smaller fat stores upon a high-fat diet [60,61]. Recently, it has been reported that the overexpression of PPARα/RXRα and PPARγ/RXRα heterodimers enhances CIC promoter activity in BRL-3A and 3T3-L1 cells, respectively [45].

## 5. Role of *DN*L Enzymes in Breast Cancer

Oncogenic signaling has been reported to increase *DN*L in order to prepare the cell for invasion and metastasis. However, it appears that not all breast cancer subtypes depend on *DN*L for fatty acid supply. Indeed, while the luminal subtypes rely on *DN*L, the TNBC subtype overexpresses genes involved in the utilization of exogenous-derived fatty acids, in the synthesis of triacylglycerols and lipid droplets, and fatty acid oxidation (Figure 2).

To support the high demand of acetyl-CoA for the increased *DN*L, luminal breast cancer cells increase glucose entry and glycolytic flux [62]. By transporting citrate into the cytosol, CIC plays an important role in *DN*L. Thus, CIC inhibition can potentially limit cancer cell proliferation. Indeed, inhibition of CIC activity by BTA was reported to reduce breast xenograft tumor growth [63].

Changes in ACLY expression have been found in diverse types of tumors including breast cancer, suggesting that this enzyme plays a crucial role in cancer metabolism [64]. ACLY has been reported to have a strong expression in breast cancer tissue, with respect to adjacent normal tissues, and silencing ACLY expression in MCF-7 cell line suppressed cell viability and increased cell apoptosis [65]. Accordingly, a study reported that genetic or chemical inhibition of ACLY reduces, both in vitro and in vivo, proliferation, and tumor growth [66].

In recent work, Lucenay et al. demonstrated that cyclin E, an independent predictor of survival in patients with invasive breast cancer, upregulating ACLY activity leads to lipid droplet accumulation, a process positively correlated with tumor growth and development [67]. ACLY mRNA has been reported to be mostly expressed in the HER2-enriched subtype with respect to TNBC, linking the expression of this enzyme to the EMT process [65,66].

Several studies have highlighted the association between ACC 1/2 and FASN expression and activity with invasion, proliferation, and EMT [66,67,68,69,70]. Enhanced of both expression and activity of FASN are considered early events in breast cancer progression [71] and blocking FASN can induce antitumor effects in TNBC [68]. Additionally, inhibition of FASN by cerulenin can affect EMT [72] and reverse the hyperglycemia-induced EMT phenotype [73]. Fasnall, a selective FASN inhibitor, reduced the proliferation of breast cancer cells and modulated the lipidomic profile of these cells by increasing ceramide levels due to malonyl-CoA accumulation and consequent CPT-1 inhibition [74]. More recently, CRISPR/Cas9 knockout of *FASN* in MCF-7 cells demonstrated that FASN inhibition has a role in reducing proliferation, cell survival, cell size, cell cycle, migration, cell adhesion, and DNA replication [75].

A study conducted by Alò and collaborators demonstrated that FASN overexpression is associated with the stage of progression of breast cancer and that FASN expression can be used as a prognostic indicator for disease-free survival and overall disease survival [76]. In breast cancer stem cell sub-populations, high expression levels of ACC 1/2 and FASN have been correlated with increased cell survival and, in turn, with the formation of pre-malignant lesions [77]. Moreover, a decreased level of palmitic acid, associated with *ACC1* and *FASN* gene silencing, can induce apoptosis in human breast cancer cells [70]. Interestingly, it has been reported that breast cancer susceptibility gene 1 (BRCA1) can exert its tumor suppressor function by preventing p-ACC1 dephosphorylation and, in turn, decreasing *DN*L [78].

ACC 1/2 and FASN expression in breast cancer cells is regulated by diverse growth factors and sex hormones through their corresponding receptors such as PR, ER, androgen receptor, and HER [79]. Based on this responsiveness, lipogenic enzyme expression is associated with molecular subtypes, and then with the malignant phenotype of breast cancers. Data suggest that in breast cancer cell lines overexpressing HER2, both FASN and ACC1 levels increased compared with cells in which HER2 expression is relatively low (such as MDA-MB-231) [80]. Indeed, induction of HER2 in MDA-MB-231 cells stimulates ACC1 expression via the PI3K/Akt pathway [81]. FASN upregulation in HER2-positive cells occurs throughout an SREBP-1-mediated mechanism. More recently, HER2 has also been shown to directly phosphorylate and activate FASN activity [82].

It must be pointed out that a metabolic transition that suppresses lipogenesis and promotes energy production is an essential component of metastasis in breast cancer. Indeed, Snail, a key inducer of EMT, has been related to ACC2 suppression and increased oxidation of mitochondrial fatty acids [83] and TGFβ1, which induces EMT, suppresses ACC in MCF-7 cells [84]. Furthermore, epithelial breast cancer cells with high expression of E-cadherin showed high expression of FASN, while mesenchymal cells with high expression of vimentin showed high expression of carnitine palmitoyltransferase-1 and therefore of β-oxidation [84]. A recent work reports that in both human and murine breast cancers, ACC1 inhibition, by increasing the level of acetyl-CoA, can favor acetylation and activation of the transcription factor Smad2, and thus EMT and metastasis [85].

It has been suggested that SCD1 may play a key role in the generation of the malignant phenotype as well as in the subsequent proliferation and survival of cancer cells [86]. Accordingly, SCD1 expression is enhanced in breast cancer tissues in situ compared to normal tissue [87,88] and SCD1 expression was associated with shorter survival times in breast cancer patients [89]. SCD1 was reported to be overexpressed in both HER2-enriched subtype [90,91] and in breast cancer cells that overexpress mucin-1 [92]. Inhibition of SCD1 activity or silencing its expression leads to anti-proliferation effects in breast cancer cell lines [93,94,95,96,97,98]. Moreover, ERα regulates SCD1 expression. Indeed, in vitro treatment of MCF-7 and T47D cell lines with 17β-estradiol induces SCD1 expression and modulates the cellular monounsaturated/saturated fatty acid ratio [99]. This was also observed in vivo, where the relative amounts of phosphatidylcholines (PC) (36:1) compared to PC (36:0) and that of PC (36:1) compared to lysoPC (18:0) were significantly higher in the cancerous areas characterized by higher levels of SCD1 expression compared to normal areas [100].

## 6. Role of *DN*L Transcription Factors in Breast Cancer

SREBP-1 has been demonstrated to play a pivotal role in breast cancer tumorigenesis, in terms of cell migration and invasion and as a prognostic marker of tumor malignancy [101]. Ectopic expression of SREBP-1 in MCF-10A cells significantly increased proliferation rate and mammosphere formation, suggesting that lipogenesis can augment the self-renewal property of these cells, thus providing oncogenic transforming abilities [77].

After MCF-7 cell exposure to MAPK and PI3K inhibitors, a reduced mRNA level for both SREBP-1c and FASN was found, while no changes in SREBP-1a and SREBP-2 levels were observed [102]. Data from Freed-Pastor and collaborators showed that fatostatin, a novel SREBP-1 inhibitor, significantly suppressed tumor growth in breast cancer [103].

ChREBP can activate target genes favoring downstream tumorigenic pathways [52]. Experimental evidence indicates that ChREBP may have a role in cancer pathology and tumorigenesis, in particular, in transformed cells that reprogram their metabolism in favor of aerobic glycolysis [104]. In this context, a study reports that ChREBP may play a key role in the malignant progression of breast cancer by allowing metabolic adaptations to take place in response to changes in oxygenation [52].

Some studies have demonstrated that PPARγ is overexpressed in breast cancer, suggesting a possible role in tumor development and/or progression [105]. An immunohistochemistry study conducted on 170 infiltrative breast carcinomas revealed that PPARγ was inversely correlated with histological grade, indicating a favorable impact of PPARγ expression on disease-free survival of patients with ductal breast carcinoma. Probable cooperation with ERβ in exerting that favorable effect was also demonstrated [106]. Moreover, selective antagonism of PPARγ with T0070907 inhibited proliferation, invasion, and migration in MDA-MB-231 and MCF-7 breast cancer cells [107].

MYC (proto-oncogene, bHLH transcription factor) is a transcription factor that controls a variety of normal functions spanning from cell cycle, cell growth, protein synthesis, mitochondrial function, and metabolism [108]. Upregulation of c-MYC and its downstream effectors is associated with poor disease outcome, high metastatic capacity, and endocrine resistance in breast tumors [109]. In luminal models, MYC appears to be involved in the regulation of metabolic pathways downstream of β-catenin and ERα signaling, thus associating the metabolic characteristics of intrinsic breast cancer subtypes on their ER expression. In particular, in breast cancer, c-MYC represents a direct target and coregulatory of ERα [110], and ERα and c-MYC act synergically to induce cell proliferation [111,112]. It has been demonstrated that β-catenin knockout reduces c-MYC expression and increases fatty acid synthesis in the breast cancer cell model MCF-7 [113].

A recent study analyzing over 2000 breast tumors highlighted the functional role of MYC in the context of TNBC. *MYC* gene amplification is associated with the risk of relapse, poor prognosis, and death. Recent studies have suggested that TNBC cells with high expression of MYC have increased fatty acid β-oxidation to support their growth. Therefore, fatty acid oxidation inhibition could be a potential therapeutic strategy for MYC overexpressing TNBC tumors [36].

N-myc downstream-regulated gene 1 (NDRG1), also called differentiation-related gene-1 (Drg1) and Cap43, is expressed in various normal tissues and suppressed in several malignancies. NDRG1 has been reported to regulate the fate of lipid in cells with altered lipid metabolism, thus contributing to breast cancer aggressiveness [114]. In vitro and in vivo data demonstrated a possible role of NDRG1 in breast cancer differentiation. NDRG1 silencing reduces cell proliferation rates, causing lipid metabolism dysfunction including increased fatty acid incorporation into neutral lipids and lipid droplets [115].

## 7. Tumor-Derived Extracellular Vesicles (EVs) Modulate Breast Cancer Metabolism

Extracellular vesicles (EVs) are cell-derived vesicles produced from likely all cell types during physiological and pathological processes [116,117,118,119,120,121]. EVs are classified in exosomes, microvesicles, and apoptotic bodies, depending on their size and biogenesis. However, the International Society for Extracellular Vesicles (ISEV), in a recent position paper, has recommended the use of the term “extracellular vesicle” for all EV types, with a generic definition in “small EVs”, if within 100–200 nm, and “medium/large EVs”, if above 200 nm [118]. EVs carry specific cargoes including genetic material (e.g., mRNAs, miRNAs, lncRNAs, nuclear, and mtDNA), proteins (e.g., cytokines, chemokines, growth factors, or other signal transduction mediators), lipids and lipid mediators (e.g., phospholipids, sphingolipids, eicosanoids) and, in the case of larger vesicles, also whole organelles (e.g., mitochondria) [118,119,121,122]. During the last years, the interest in these vesicles has grown considerably due to their crucial role in many pathological conditions including autoimmune, inflammatory, cardiovascular, metabolic diseases and tumors [118,121,123,124,125].

In breast cancer, EVs are characterized by a specific protein cargo useful for the detection and classification of breast cancer subtypes [125]. This includes proteins, phosphoproteins, protein kinases, and metabolic enzymes. For instance, glycolytic enzymes were identified in breast cancer EVs including aldolase, glyceraldehyde 3-phosphate dehydrogenase, enolase, triosephosphate isomerase, fructose bisphosphatase 1, and phosphoglycerate kinase [125,126]. Santi and collaborators demonstrated that cancer-associated fibroblasts can support tumor growth through the transfer of lipids and proteins to cancer cells by EVs [127]. Moreover, Achreja and colleagues demonstrated that cancer cells internalize cancer-associated fibroblast EVs rapidly and that this phenomenon influences intracellular metabolism. Indeed, lactate, transported by cancer-associated fibroblast EVs, regulates glycolysis flux, and contribute up to 35% of the TCA cycle flux by providing TCA intermediates and glutamine [128]. Sansone and colleagues observed that the full mitochondrial genome was packaged in cancer-associated fibroblast-derived EVs and EVs isolated from patients with hormonal therapy-resistant breast metastatic disease. The acquisition of cancer-associated fibroblast-derived EVs-mtDNA by breast cancer cells influences metabolism, promoting estrogen receptor-independent oxidative phosphorylation [129].

In a metabolomic analysis of cancer-associated fibroblast-derived vesicles, it was found that EVs can transport metabolites required for lipid synthesis such as acetate, which is required for lipid synthesis [130]. These data suggest that EVs can supply recipient cells with lipogenic substrates, a feature highly relevant in cancer where tumor cells need these building blocks to proliferate.

Recent studies have demonstrated that extracellular vesicles are able of carrying lipids from parent cells to recipient cells such as fatty acids, cholesterol, eicosanoids, etc. which may cause, among other things, metabolic changes [131,132,133]. However, a growing number of reports provide evidence that extracellular vesicles can regulate the expression of classical lipid transporters such as CD36, ATP-binding cassette transporter A1 (ABCA1), low-density lipoprotein receptor (LDLR), and ATP-Binding Cassette Subfamily G Member 1 (ABCG1) [134]. A targeted quantitative lipidomic analysis of EVs and cells derived from high-metastatic (D3H2LN) and low-metastatic (D3H1) TNBC cell lines found that unsaturated diacylglycerol species were upregulated in EVs from high-metastatic D3H2LN cells when compared with low-metastatic D3H1 cells without an increase in secreting cells. EVs enriched in unsaturated diacylglycerols can induce phosphorylation of PKD/PKCμ and PKCδ in endothelial cells, which leads to stimulation of neo-angiogenesis. Moreover, unsaturated EV-derived diacylglycerols may contribute to EV-mediated education of other surrounding cells to support tumor progression [135].

However, the role of EVs on lipogenesis depends on the cell of origin and growth conditions. miR-126-3p was found to decrease lipid accumulation in mammary epithelial cells, while its inhibition led to an increased number of intracellular lipid droplets concomitantly with an upregulation, among others, of FASN and acyl-CoA synthetase long-chain family member 1 (ACSL1) levels [136].

EVs are emerging as a novel mechanism to allow fatty acid transport between cells and across cell membranes [133]. Indeed, FABPs, key extracellular and intracellular fatty acid transporters, were found abundant in EVs released from many cell types [133].

Although the exact function within breast cancer is not yet fully understood, all of these data suggest that EVs could play important roles in influencing lipid metabolism in breast cancer. Therefore, the knowledge of their role in the different subtypes of breast cancer could open interesting fields of study.

## 8. Conclusions

Collectively, different studies indicate that breast cancer subgroups have a specific lipogenic phenotype that can support a different metabolic demand, providing a metabolic fingerprinting useful to classify cancer subtypes [137]. Therefore, while luminal subtypes upregulate *DN*L, the basal-like model relies on uptake and storage of exogenous fatty acids, which ultimately direct to β-oxidation. In this complex scenario, extracellular vesicles seem to have an important role in tumor metabolic reprogramming, allowing the exchange, between cells, of enzymes and metabolites useful for the lipogenic process. However, the understanding of the mechanism at the basis of these cell communications still requires considerable in-depth analysis.

Exogenous fatty acids might contribute to the constitution of structural lipids such as sphingolipids, phospholipids, and cholesterol, and non-structural lipids such as triacylglycerols and cholesteryl esters. Based on these considerations, it can be expected that lipogenic differences among breast cancer subtypes can differently affect membrane-associated cellular processes such as vesical trafficking, signal transduction, and molecular transport [138]. Indeed, the endoplasmic reticulum membrane architecture and associated enzymatic activities were mostly influenced in TNBC compared to the luminal cell line after a long exposure to exogenous added polyunsaturated fatty acids [139].

On the other hand, cells characterized by enhanced *DN*L can present membranes enriched with saturated and/or monounsaturated fatty acids, the end product of *DN*L and SCD1 activity [140]. These cells are less prone to lipid peroxidation than cells with more unsaturated membranes and more resistant to peroxidative damage and cell death [140,141]. Moreover, as saturated lipids pack more densely, increased *DN*L can also alter lateral and transverse membrane dynamics that may limit the uptake of drugs, making the cancer cell more resistant to therapy [139].

In TNBC, the described lipid metabolic remodeling can sustain the activation of specific cellular processes or/and modify the lipidomic composition of these cells.

For instance, it appears that the activation of the EMT program is associated with an increased expression of fatty acid uptake proteins including CD36 [98], and exogenous fatty acids such as linoleic and arachidonic acid can initiate EMT in the human breast epithelial cell line MCF-10A [141,142]. The dependence of TNBC from exogenous fatty acids is reflected in the higher amount of phospholipid enriched in fatty acid with double bonds in position 11 (PC 34:1) in TNBC with respect to the ER/PR+ and HER2+ subtypes [143]. Moreover, the prevalent incorporation of exogenous fatty acids in basal-like breast cancers, with respect to the estrogen-positive subtypes, is reflected in an increased level of polyunsaturated fatty acid-enriched PC and cardiolipin molecules [141]. In this scenario, the increased propensity of TNBC to form lipid droplets is seen as a way to sequestrate potentially toxic lipids to maintain survival [144]. This aspect of TNBC cells makes this cancer subtype more vulnerable to intervention toward the pathway of lipid droplet formation. Additionally, knowing that claudin-low TNBC patients have a strong dependence on fatty acid import throughout CD36, possible interventions toward CD36 and/or fatty acid oxidation pathway in claudin-low TNBC patients have been proposed [36].

Overall, we can conclude that lipid metabolism represents an attractive model for anticancer drug studies since it not only differs between normal tissues and tumors, but also varies between tumor subtypes and concerning malignancy. Differences in the lipid metabolism of breast cancer subtypes prompt efforts to uncover disease-specific lipid alterations that can be proposed as diagnostic biomarkers.

## Figures and Tables

**Figure 1 ijerph-18-03575-f001:**
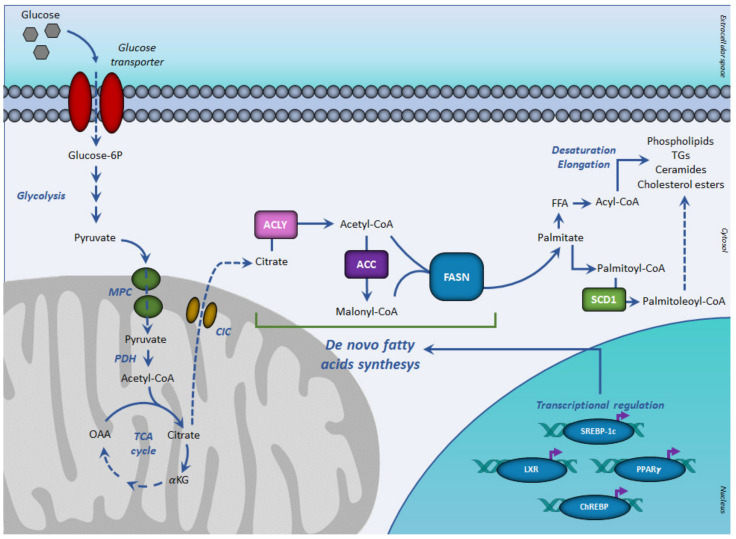
De novo lipogenesis and transcriptional regulation. In aerobic conditions, glucose-derived pyruvate fuels, in the form of acetyl-CoA, the tricarboxylic acid cycle, to form citrate. Once exported in the cytosol, citrate generates acetyl-CoA for fatty acid synthesis, mediated by sequential reactions of acetyl-CoA carboxylase and fatty acid synthase. The resulting palmitoyl-CoA is used for the synthesis of complex lipids. The de novo fatty acids synthesis is regulated at the transcriptional level by the SREBP-1c, PPARγ, ChREBP, and LXR receptor family. Abbreviations: ACC, acetyl-CoA carboxylase; ACLY, ATP citrate lyase; ChREBP, carbohydrate-responsive element-binding protein; CIC, citrate carrier; FASN, fatty acid synthase; FFA, free (non-esterified) fatty acid; αKG, α-ketoglutarate; LXR, liver X receptor; MPC, mitochondrial pyruvate carrier; OAA, oxaloacetate; PPARγ, peroxisome proliferator-activated receptor-γ; PDH, pyruvate dehydrogenase; SCD1, stearoyl-CoA desaturase; SREBP-1c, sterol regulatory element-binding proteins-1c; TCA, tricarboxylic acid cycle.

**Figure 2 ijerph-18-03575-f002:**
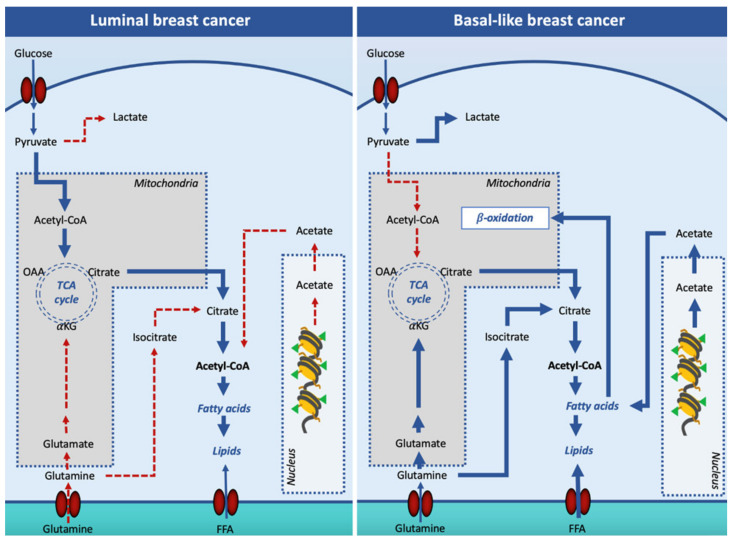
Lipogenesis in luminal and basal-like breast cancer cells. In luminal breast cancer cells, glucose-derived acetyl-CoA is the main source of citrate for the cytosolic synthesis of lipids. In basal-like, pyruvate is mainly converted into lactate in the cytosol, and other substrates such as glutamine and acetate are used to support cell lipid synthesis. Additionally, basal-like breast cancer cells increase free fatty acid entry in the cell to fulfill the β-oxidation pathway. Solid arrows signify the main reaction processes and dotted arrows signify processes with a minor relevance. Abbreviations: FFA, free (non-esterified) fatty acid; OAA, oxaloacetate; αKG, α-ketoglutarate; TCA, tricarboxylic acid cycle.

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
