# Peer review of "Expanding Roles of De Novo Lipogenesis in Breast Cancer"

_ijerph, 2021, doi:10.3390/ijerph18073575_

Round 1

Reviewer 1 Report

The authors aim with this review expand in a revised format de roles of the de novo lipogenesis in breast cancer. 

After carefful reading the proposed manuscript in my opinion it needs major improvements and rearrangements, since I detected several misleading and not perfectly clear ideas and statements.

The authors used a single figure for the manuscript that has a very extensive description in the legend and is not well connected to the subparts of the paper. It seems the figure is completely disconnected from the rest of the text.

Although it is a review on de novo lipogenesis, the authors should in my opinnion have started by describing the major diference that exists between lipogenesis and the de novo lipogenesis and the role of cancer in this diference. In normal tissues, lipids come from circulating lipids while cancer cells use the de novo synthesized lipids. As such, the rate of lipogenesis is higly induced in cancer. This simple distinction in the field is not explained in this review and could be used to highlight the great value of de different metabolic status of cancer cells

Also, detected some wrong and misleading statements relativy to the process of breast cancer progression and classification. The statement for instance in line 385 that MCF10A is a breast cancer cell line is completely wrong. MCF10A is  non tumorigenic cell line from the breast epithelium. With so many abreviations included throuhout the text, for instance, is not explained what TNBC stands for. And this is a subtype classification of breast cancer. 

Section 6 also seems to have some confusion since it starts to describe Cancer associated fibroblasts in the contex of EVs and lipids and tumor growth (line 336) but then in line 362 re-starts again with more information on this theme. 

In section 7, the authors conclude that overall the lipid metabolism represents an attractive target for anti-cancer drugs since differs between normal tissues and tumors and between tumor subtypes and malignancy . With this review, I do not see how the authors can take such a conclusion given the works and referenced results that were presented in the previous sections 

In conclusion, I believe the paper in the present format is not well conducted, with a clear improvement to the field and for that I do not recomend its publication.

Author Response

Dear Reviewer,

We thank you both for your valuable time spent evaluating our manuscript and we thank you for the nice comments and precious suggestions you made, which helped us to improve our manuscript. We now prepared a revised version of our manuscript, which incorporates the changes you suggested. We hope that in the present form the manuscript will be suitable for publication.

Reviewer 1.

The authors aim with this review expand in a revised format de roles of the de novo lipogenesis in breast cancer. 

After carefful reading the proposed manuscript in my opinion it needs major improvements and rearrangements, since I detected several misleading and not perfectly clear ideas and statements.

  • The authors used a single figure for the manuscript that has a very extensive description in the legend and is not well connected to the subparts of the paper. It seems the figure is completely disconnected from the rest of the text.

We thank the reviewer for the suggestion. According, we corrected Figure 1 leaving only the main enzymes of the de novo fatty acid synthesis pathway (by eliminating the part concerning the entry of fatty acid in the cell), and a more easy legend. Moreover, we also added a new Figure 2 where differences between luminal and basal-like breast cancer lipogenic phenotypes are better highlighted.

  • Although it is a review on de novo lipogenesis, the authors should in my opinnion have started by describing the major diference that exists between lipogenesis and the de novo lipogenesis and the role of cancer in this diference. In normal tissues, lipids come from circulating lipids while cancer cells use the de novo synthesized lipids. As such, the rate of lipogenesis is higly induced in cancer. This simple distinction in the field is not explained in this review and could be used to highlight the great value of de different metabolic status of cancer cells.

We agree with the reviewer about this point. According, we inserted a new paragraph where we indicated the main differences between de novo lipogenesis and lipogenesis and their different contribution to breast cancer.

  • Also, detected some wrong and misleading statements relativy to the process of breast cancer progression and classification. The statement for instance in line 385 that MCF10A is a breast cancer cell line is completely wrong. MCF10A is  non tumorigenic cell line from the breast epithelium. With so many abreviations included throuhout the text, for instance, is not explained what TNBC stands for. And this is a subtype classification of breast cancer. 

We apologize for the mistake. We corrected the wrong sentence.

  • Section 6 also seems to have some confusion since it starts to describe Cancer associated fibroblasts in the contex of EVs and lipids and tumor growth (line 336) but then in line 362 re-starts again with more information on this theme. 

According to the reviewer's suggestion, we have better organized section 6 related to extracellular vesicles.

  • In section 7, the authors conclude that overall the lipid metabolism represents an attractive target for anti-cancer drugs since differs between normal tissues and tumors and between tumor subtypes and malignancy . With this review, I do not see how the authors can take such a conclusion given the works and referenced results that were presented in the previous sections.

According to the reviewer’s suggestion, we thoroughly rearranged the discussion section, emphasizing how the different lipogenic phenotypes of breast cancer subtypes could influence cell behaviors.

Reviewer 2 Report

This manuscript summarized the current status of research directed at understanding the roles of de nove lipogenesis and the relationship of this processing and breast cancer subtypes. This manuscript is enjoyable to read and conclusions are well supported by their provided evidence and references.
Minors: 

  • Line 207: Lucenay et coll --> et al.
  • Line 261: Breast cancer lines --> Breast cancer cell lines; in vitro in this line is not italic

Author Response

Dear Reviewer,

We thank you both for your valuable time spent evaluating our manuscript and we thank you for the nice comments and precious suggestions you made, which helped us to improve our manuscript. We now prepared a revised version of our manuscript, which incorporates the changes you suggested.

Reviewer 2.

This manuscript summarized the current status of research directed at understanding the roles of de nove lipogenesis and the relationship of this processing and breast cancer subtypes. This manuscript is enjoyable to read and conclusions are well supported by their provided evidence and references.
Minors: 

  • Line 207: Lucenay et coll --> et al.

Done

  • Line 261: Breast cancer lines --> Breast cancer cell lines; in vitroin this line is not italic

Done

Reviewer 3 Report

The present manuscript entitled “Expanding roles of de novo lipogenesis in breast cancer” focuses on de novo lipogenesis (DNL) and the specific differences exhibited by breast cancer subtypes. This manuscript also examines the functional contribution of lipogenic enzymes and associated transcription factors, in the regulation of tumorigenic processes.

I have the following comments.

This manuscript resembles like a collection of literature. It lacks the opinion of the authors. I would like to suggest the authors to improve the conclusion and abstract section.  

I would like to suggest the authors to provide a figure summarizing the role of DNL in breast cancer subtypes as a graphical abstract.

The authors quote “luminal subtypes upregulate DNL, the basal-like model relies on uptake and storage of exogenous fatty acids which ultimately direct to β-oxidation”. So, during the process of exploitation of exogenous fatty acids, there occurs the generation of multiple signaling lipids. The authors should also discus on signaling lipids and basal-like breast cancer.

Typo errors in line 400 and 401.

Author Response

Dear Reviewer,

We thank you both for your valuable time spent evaluating our manuscript and we thank you for the nice comments and precious suggestions you made, which helped us to improve our manuscript. We now prepared a revised version of our manuscript, which incorporates the changes you suggested. We hope that in the present form the manuscript will be suitable for publication.

Reviewer 3.

The present manuscript entitled “Expanding roles of de novo lipogenesis in breast cancer” focuses on de novo lipogenesis (DNL) and the specific differences exhibited by breast cancer subtypes. This manuscript also examines the functional contribution of lipogenic enzymes and associated transcription factors, in the regulation of tumorigenic processes.

I have the following comments.

  • This manuscript resembles like a collection of literature. It lacks the opinion of the authors. I would like to suggest the authors to improve the conclusion and abstract section.

According to the reviewer indication, we rearranged the discussion section to better highlight how a different lipogenic phenotype can induce membrane-associated typical pathways of different breast cancer.

  • I would like to suggest the authors to provide a figure summarizing the role of DNL in breast cancer subtypes as a graphical abstract.

We thank the reviewer for the suggestion. Accordingly, we made a new Figure 2 that we propose as a graphical abstract.

  • The authors quote “luminal subtypes upregulate DNL, the basal-like model relies on uptake and storage of exogenous fatty acids which ultimately direct to β-oxidation”. So, during the process of exploitation of exogenous fatty acids, there occurs the generation of multiple signaling lipids. The authors should also discus on signaling lipids and basal-like breast cancer.

We agree with the reviewer about the importance of this topic. Accordingly, we better discussed this aspect in the new version of the manuscript.

  • Typo errors in line 400 and 401.

Done